# Academic Stress in University Students: The Role of Physical Exercise and Nutrition

**DOI:** 10.3390/healthcare11172401

**Published:** 2023-08-26

**Authors:** Montserrat Monserrat-Hernández, Juan Carlos Checa-Olmos, Ángeles Arjona-Garrido, Remedios López-Liria, Patricia Rocamora-Pérez

**Affiliations:** 1Social Anthropology, Department of Geography, History and Humanities, University of Almeria, Carretera de Sacramento S/N, La Cañada de San Urbano, 04120 Almería, Spain; mmh548@ual.es (M.M.-H.); arjona@ual.es (Á.A.-G.); 2Sociology, Department of Geography, History and Humanities, University of Almeria, Carretera de Sacramento S/N, La Cañada de San Urbano, 04120 Almería, Spain; jcheca@ual.es; 3Health Research Centre, Department of Nursing, Physiotherapy and Medicine, University of Almeria, Carretera de Sacramento S/N, La Cañada de San Urbano, 04120 Almería, Spain; rll040@ual.es

**Keywords:** diet, Mediterranean, physical activity, stress

## Abstract

The university experience can cause academic stress that, in turn, can lead to comorbidities. Students increasingly face demands and challenges that require a large amount of physical and psychological resources. These circumstances can make them experience physical and mental fatigue, lower their interest in studying, and even lead them to lose control over their academic performance and health. The objective of this work is to determine the relationship between the practice of physical exercise, eating patterns, and academic stress among university students. A questionnaire was administered to 742 students using non-probabilistic sampling. The mean age was 21.24 (DT = 3.8), and 20.1% were men and 79.4% were women. To identify academic stress, the Stress Manifestation Scale of the Students Stress Inventory (SSI) subscale was used; the Mediterranean Diet Score was applied for eating patterns, and the practice of exercise was measured by weekly hours of exercise. The results show that there is a relationship between academic stress and physical exercise, but not with adherence to the Mediterranean Diet. However, there is a relationship between the consumption of “unhealthy” foods exceeding the recommendations for the Spanish population and academic stress. In short, physical activity and diet are variables that are related to psychological well-being. Therefore, they should constitute the backbone of actions designed by university managers to eliminate or reduce stress suffered by students. Finally, the work demonstrates the need to create new scales that consider not only the foods that help alleviate stress but also their portions.

## 1. Introduction

Stress is a very broad concept, encompassing various fields. Not only is it associated with both psychological and physical health, but it is also related to social circumstances, subjective perspective, and interactions with the environment [1,2]. In this regard, the university experience constitutes a life stage carrying significant stress, also known as academic stress, understood as a physiological, emotional, cognitive, and behavioral activation reaction to stimuli and academic events [3]. This stress can be beneficial as long as it is controlled and helps keep the student alert to situations that require so [4], but when these situations are maintained for too long or generate harmful physiological and psychological responses, distress appears [5], which may lead the student to consider as anguished the vast majority of the situations they have to face [5,6].

This problem, which is currently growing so much internationally [7,8], is being widely investigated, especially with studies of prevalence [9] and characterization of the phenomenon, analyzing psychological and academic variables [10,11], considering them as causal and/or modulatory variables for the presence of the phenomenon [12,13,14]. Also, studies carried out during the COVID-19 pandemic and the obligation of distance learning [15,16,17] revealed the presence of higher levels of stress compared to the pre-pandemic era, highlighting the importance of personal relationships.

In general, research points to the moment of transition in which young people find themselves as the main triggering factor, in which important hormonal, cognitive, and emotional changes occur [18]. In the same way, the university student is exposed to a new and more demanding academic context that redefines, in turn, relationships with peers [19,20], as well as romantic relationships [21], expectations about the future, which are often confusing [22], economic pressures [23] and new norms about body image [24].

In addition, it has been observed that in the university stage, there is a loss of healthy habits among university students compared to previous stages [25,26]. It seems that the need to reorganize time can affect leisure, physical exercise, social relationships, and even eating habits [27,28,29]. The scientific literature shows that during exam time (a stressful situation), students modify their eating and physical exercise habits [30,31,32,33]. On the one hand, they increase their consumption of sugary food high in saturated fat, salt, and calories [34,35,36], or they drastically reduce their intake because they feel repulsed by food [27]. And regarding physical exercise, while it decreases in some cases, it increases in others [37].

The vast majority of previous studies on stress in students are based mainly on analyses of prevalence (situation and population affected), descriptive (what they do), and causal or consequential (causes for which it appears and associated consequences) analysis. On the other hand, it has been observed that the practice of healthy habits can manage stressful situations. Jimenez et al. [38] show that, regardless of the type of exercise (aerobic, anaerobic, or strength), it always provides benefits. Other research affirms that an organized, detailed, and healthy exercise regime produces positive psychological effects for the individual, such as the prevention of stress, since it stimulates the feeling of competence and is established as a means to increase self-control and self-sufficiency, in addition to providing a time to avoid unpleasant thoughts, emotions and behaviors [39,40].

Regarding eating patterns, there is abundant literature that demonstrates the benefits of the Mediterranean Diet (MD) on stress [41,42,43]. In general, it is evident that MD can combat stress thanks to its contribution of macro and micronutrients in the appropriate proportion, helping to maintain a physically and psychologically healthy body [43,44,45,46,47].

Therefore, it has been possible to observe how stress and health (both physical and psychological) interact [48,49]. At the level of the young population, the literature shows that the university period and especially the moments of exams influence eating habits and physical exercise, but there is not enough research to show the inverse relationship (that is, physical exercise and eating habits influence the perception of stress). For this reason, the main objective of this research has been to analyze the relationship between eating patterns and the practice of physical exercise with the perception of academic stress.

Furthermore, when the level of adherence with the MD is measured, as well as when other measurement scales are used, such as the Healthy Eating Index [47], to compare it with stress levels, it does not take into account the possibility of calculating the consumption of certain foods and nutrients (beneficial or harmful), which have been shown to have an impact on the nervous system [50,51,52,53,54]. For this reason, our second objective has been to find out if the consumption of certain foods is related to the stress suffered by young students.

## 2. Material and Methods

This work is part of the Healthy and Sustainable Eating Program at the University of Almeria 2020–2022 (ASASO-UAL), developed within the Office of the Vice President for Sports, Sustainability and Healthy University of the University of Almeria.

### 2.1. Participants

The sample consisted of a total of 742 students selected by non-probabilistic sampling (149 men (20.1%) and 589 women (79.4%)). The mean age was 21.24 (SD = 3.8), 86.4% of participants had Spanish nationality, and 13.6% were foreign.

### 2.2. Instrument and Procedure

Data collection was carried out using an anonymous questionnaire through the Google Forms platform, applied to university students in the month of October 2022. This document collected sociodemographic data (sex, age, country of birth), frequency of consumption of food, weekly hours of physical activity, and manifestations of academic stress.

Measuring the frequency of food consumption was carried out using the Mediterranean Diet Score (MDS) with the modifications made by Schröder et al. [53]; unlike the original [54], item 8, related to wine consumption, is eliminated, reducing it from 14 to 13 items. Scores range from 0 to 13, where less than 8 points means “low adherence”, between 8 and 10 points “good adherence”, and more than 10 points “very good adherence”. In addition, to achieve the objective of the study of relating food consumption to stress, other foods not included in the MDS were added to the questionnaire but for which there is scientific evidence of their relationship with the manifestation of stress, both positive and negative, such as the consumption of blue fish [55].

Academic stress was measured through the Stress Manifestation Scale of the Students Stress Inventory (SSI) [56], which comprises four independent self-report subscales and assesses four different elements: physical, interpersonal relationships, academic, and environmental. In this study, only the third subscale adapted to Spanish (academic stress) has been used. It consists of the items: (1) I have financial problems due to university expenses; (2) I find it difficult to reconcile study time with social activity; (3) I feel nervous when I have to present in class; (4) I feel stressed by the deadline for the delivery of work; (5) I feel stressed by exams; (6) I find it difficult to juggle study time and my leisure activities; (7) I feel overwhelmed by academic loads; (8) I loss interest in the course I take; (9) I feel stressed when something is difficult; (10) I have difficulty in handling academic problems.

The scale ranges from 0 to 10 points, where values between 0 and 3 show a low level of stress (high performance, high motivation, and good time management), between 4 and 7 a medium level of stress, and between 8 and 10 show severe stress (very low performance, low self-motivation, and lack of time). In the present study, to improve the analysis of results, the scale was reorganized from 0 to 5: “no stress”, “very low stress”, “low stress”, “medium stress”, “high stress”, and “very high stress”.

The practice of exercise was measured with the question: How many hours do you do physical exercise per week?

### 2.3. Statistic Analysis

Descriptive analyses regarding adherence to the MD, physical exercise, and manifestation of academic stress were performed for our study sample. Subsequently, we proceeded to verify whether there were interactions between the practice of physical exercise, adherence to the MD, and academic stress while considering our hypothesis. In light of the perception of stress, there are differences between people who practice physical exercise and eat according to recommended standards and those who do not.

Because the test for normality (Kolmogorov–Smirnov) for the three variables showed that the distribution was not normal, non-parametric tests were performed for independent samples. In general terms, it can be considered that, although the power of the parametric statistical tests is greater than the non-parametric ones, it is worth mentioning that the adequate size of the sample is an essential requirement to increase the effectiveness of the test, as the sample size decreases the possibility of committing the type II error [57], and in our case, the sample is made up of 742 subjects. For this reason, the Kruskal–Wallis test was carried out, with which we assume that: (1) the data come from a random group of observations; (2) the dependent variable is ordinal; (3) the independent variables are nominal; (4) the observations are independent within each group and between them; and, (5) there are no repeated measures or multiple response categories. The post-hoc tests carried out (Bonferroni) showed that there were no significant statistics when adherence to the Mediterranean Diet was used as the reference variable, while significant differences (*p* < 0.001) were observed between the related pairs when using the practice of physical exercise as the reference variable. The effect size was performed with physical exercise as an independent variable through Hedges’ g because it was the only one that showed significance.

Subsequently, bidirectional correlations were performed when analyzing the influence of the consumption of certain foods (some not included in the MDS) on the stress levels of the student body. However, since the number of foods considered was very high, and there was a high correlation between some of them, a factor analysis with the maximum likelihood method (extraction method: principal component analysis, Varimax rotation with Kaiser normalization) was carried out to reduce dimensionality and avoid multicollinearity problems (components that saturate at values less than 0.300 were excluded). For the correlation, the factors with high scores extracted from the factor analysis were used.

The data were analyzed with the statistical program SPSS-27 for Windows.

## 3. Results

### 3.1. Adherence to the Mediterranean Diet, Stress Level, and Sports Practice of University Students

As can be seen in Table 1, only 20 people (2.7% of women and 2.6% of men) show high adherence to the MD. None achieved the maximum score (13 points), and 511 students reported low adherence: 68.2% of women and 73% of men. Nevertheless, there are no significant differences between men and women (Chi-square = 2.196; *p* = 0.526).

Regarding the practice of physical exercise, 65.5% of women and 49.7% of men do less than 4 h a week; more specifically, women show a greater sedentary lifestyle than men. Therefore, and for the opposite case, 50.3% of men and 34.4% of women exercise more than 4 h a week. In this aspect, significant differences are observed (Chi-square = 20.478) (*p* ≤ 0.001).

Furthermore, regarding the manifestation of academic stress, it is observed that 57 university students (8.3% of women and 5.3% of men) are at high or very high levels, and 106 are at medium levels (14.6% of women and 12% of men). There are no significant differences between men and women (Chi-square = 7.817; *p* = 0.647).

### 3.2. Relationship between the Practice of Physical Exercise, Adherence to the MD, and Academic Stress

Based on the non-significant Chi-square between gender and academic stress, we performed the Kruskal–Wallis test for independent samples, both for the practice of exercise and for adherence to the MD in relation to academic stress. We accept the null hypothesis (*p* = 0.244) for adherence to the MD, which is to say the distribution of MDS is the same between the categories of feeling stress or anxiety. Nonetheless, we also accept the alternative hypothesis for exercise (*p* < 0.001) since the distribution of exercise is not the same between categories of feeling stress or anxiety. Therefore, with regard to the practice of physical exercise, there are significant relationships in the test overall since the contrast statistic was 50.639 in the Kruskal–Wallis test, obtaining an Asymptotic sign (two-sided test) of *p* < 0.001.

More specifically, Table 2 (corresponding Figure 1) shows exactly which pairs significant contrast levels occur. The highest and most significant contrast data are observed between “high stress” and “very low stress”; “high stress” and “no stress”; and “medium stress” and “no stress”.

Regarding the adherence to the MD, it was shown that the relationships are neither significant as a whole (contrast statistic 6.703), asymptotic significance (bilateral test *p* = 0.24), nor between pairs (Figure 2).

The size and direction of the effect shown with significant values (Hedges’ g < 0.7; *p* < 0.001) that people who perform weekly physical exercise according to the World Health Organisation (WHO) recommendations [58] show “no stress” and “very low stress”, the rest of the variables showed no significant relationship (see Table 3).

### 3.3. Academic Stress and Food Consumption

The factorial analysis reflects an optimal saturation of the items in each factor, presenting high representativeness in each of them (see Table 3). Each item shows whether the subject possesses an optimal consumption of food according to the recommendations of the Spanish Agency for Food Safety (AESAN) [59].

Thus, Factor 1, called Unhealthy Eating, consists of not consuming at least two pieces of fruit a day and consuming highly processed snacks, sugary drinks, and processed baked goods two or more times a week.

Factor 2, called Healthy Eating, is characterized by a minimum daily consumption of three tablespoons of olive oil, consumption of natural or roasted nuts three or more times a week, and consumption of oily fish two or more times a week.

Regarding the degree of determination of the factors, the Total Explained Variance (TEV) reaches 93.99%, which implies high representativeness. Therefore, in keeping with the objective of identifying what type of relationship exists between eating habits and stress, bilateral correlations are performed. As shown in Table 4 and Table 5, there is a highly significant and positive relationship between the consumption of foods considered unhealthy (AESAN), exceeding MD recommendations, and the manifestation of academic stress.

## 4. Discussion

In general, the low levels of adherence to the MD are similar to the results of recent research regarding the university population [60,61]. The MD is a prototype of a healthy diet that represents a lifestyle circumscribed to a territorial framework with a particular climate, and although these aspects remain the same, young people tend to frequent fast food establishments and/or consume highly processed foods, demonstrating a loss of this legacy [36,40,62].

The fact that all current research shows a loss of adherence to the MD by the young community reveals two lines of work for experts. Firstly, there is a need to improve nutrition education, perhaps through more influential channels for these population groups, to generate greater adherence from an early age. Secondly, the possibility of updating the scales for measuring eating habits, taking into account social, economic, and regional circumstances, should be considered.

Regarding the results of academic stress, our research does not match the findings of current studies in terms of gender [63]. There are no significant differences between men and women. Notwithstanding, the fact that the population analyzed shows mostly medium levels of academic stress is alarming since this can lead to (1) dropping out of studies [64,65]; (2) the development of behaviors that are dangerous to health, such as the use of drugs or alcohol to escape [64]; or, (3) the manifestation of psychological disorders such as depression, anxiety or generalized stress [66].

Although no significant relationship was found between adherence to the MD and academic stress in this study, there was a correlation with low fruit consumption, consumption of sugary drinks, and highly processed snacks and baked goods. In this regard, there are studies that observe high consumption of sugary and highly processed foods among young people during exam periods [34,35,36]. Even so, our research makes an additional contribution by showing that the consumption of sugary and highly processed foods can also influence the manifestation of academic stress. These results propose a novel line of work with respect to recent studies on the gut–brain axis and how what we eat affects our emotions [39,67,68].

In addition, the fact that no significant relationship has been observed regarding the consumption of foods considered “protective” by the scientific community regarding stress prevention [68] shows that common measurement methods only obtain information on generic habits (healthy or not) but not enough to analyze the nutrient–stress relationship. For example, the MDS does not distinguish between the consumption of white and blue fish [54]. For this reason, not only do we believe it is worthwhile to consider the MD as a healthy and preventive lifestyle, but also, depending on the objectives of the consumers, to know which foods can be more favorable for one’s own self-care.

Regarding exercise, as observed in previous studies [60,69], university women are more sedentary than men. An important fact to highlight is that, of the people who exercise, most do so for more than 4 h a week, which is why it is easy to generate adherence when the positive effects are observed and known. In addition, it has been observed that the practice of physical exercise on a regular basis, according to the recommendations of the WHO [58], can be of help to prevent and against stress. These positive aspects, along with the fact that a high percentage of the population analyzed presented medium levels of academic stress, reveal the need for a preventive and palliative approach.

Nevertheless, even though physical exercise is promoted on university campuses, this may not be enough since the problem may lie in a lack of time, organization, excessive workload, or even ignorance of the benefits it provides [70].

## 5. Limitations and Future Directions

The main limitations of this study include, firstly, a greater participation of women. It should be noted that this is a difficult characteristic to control in an environment where research resources are not abundant.

Secondly, the fact that a cross-sectional study was carried out limits the terms of the conclusions with respect to cause–effect relationships. In addition, the fact that the research is based on non-probabilistic sampling makes it difficult to generalize the results to the target population in general.

Thirdly, not enough variables were introduced for the study of physical activity, such as modality or intensity, which is an aspect to consider in future research.

Fourthly, the existence of a possible “floor effect” of the scale makes it advisable to use another type of scale to measure academic stress in future research.

Despite these limitations, we consider that this study offers very interesting preliminary data with high explanatory and justifying value in order to lay the foundations for future research in a broader university population and thus be able to contribute to the construction of eating and exercise patterns for young university students to cope with stress. It could also be interesting for future research to study the association between food groups and emotional state or even the function of emotional regulation through its consumption.

## 6. Conclusions

In conclusion, we believe that both the prevention of stress and the promotion of physical exercise and healthy eating should be among universities’ health initiatives and proposals, not only as optional activities but as part of the transversal contents of the curriculum in all fields of study. Perhaps even further measures could be taken to develop an interdisciplinary approach to establish a distinction regarding gender, seeking to improve adherence to sport among the female population.

In addition, having observed the results on the type of diet related to the manifestation of stress, the possibility of creating measurement scales for food groups and/or nutrients (daily amounts and proportions) should be investigated, especially focusing on the manifestation of stress.

## Figures and Tables

**Figure 1 healthcare-11-02401-f001:**
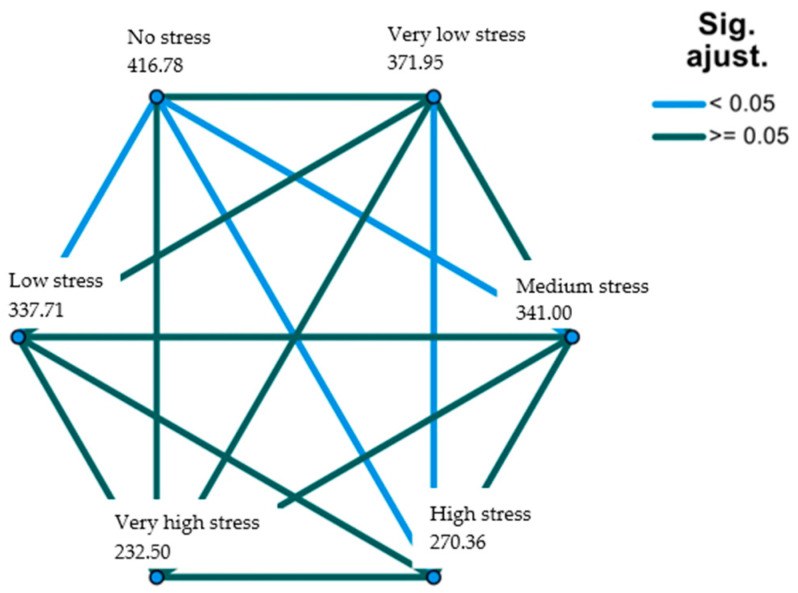
Academic stress relationships by pairs (physical exercise as a reference). Source: own elaboration.

**Figure 2 healthcare-11-02401-f002:**
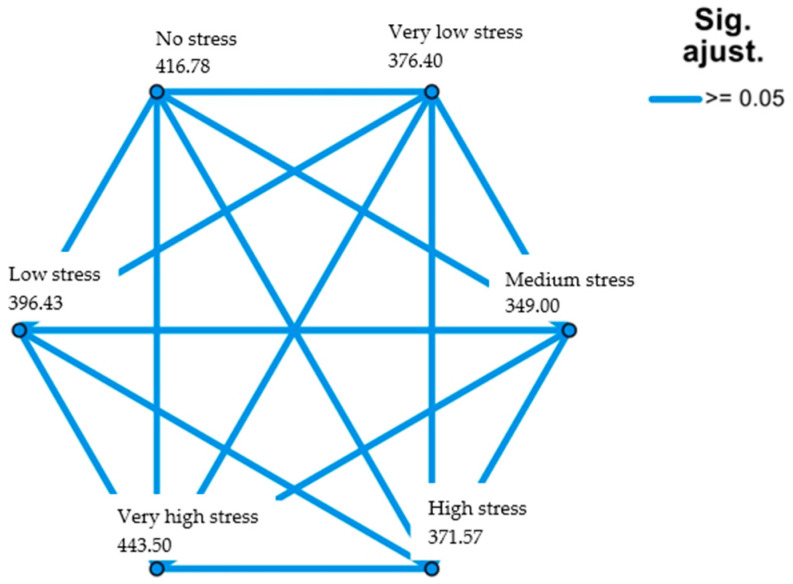
Academic stress relationships by pairs (MD adherence as reference). Source: own elaboration.

**Table 1 healthcare-11-02401-t001:** Basic data of the research screening sample.

	Mean	Standard Deviation	Range
Age	24	2.12	18–30
Hours per week dedicated to sports practice *	1.2	0.7	0–3
Level of Adherence to the MD **	0.3	0.5	0–2
Stress level ***	1.3	1.35	0–5
	Female	Male	I prefer not to say
	No. (%)	No. (%)	No. (%)
Hours of sports practice per week			
Less than 2 h	134 (22.7)	22 (14.8)	3 (75)
Between 2 and 4 h	252 (42.8)	52 (34.9)	1 (25)
More than 4 h	203 (34.4)	75 (50.3)	0 (0)
MD Adherence Level			
Low	402 (68.2)	109 (73)	4 (100)
Medium	171 (29.0)	36 (24)	0
High	16 (2.7)	4 (2.6)	0
Academic stress level			
No stress	234 (39.7)	69 (46.3)	1 (25.0)
Very low stress	109 (18.5)	32 (21.5)	0 (0)
Low stress	111 (18.8)	22 (14.7)	1 (25.0)
Medium stress	86 (14.6)	18 (12)	2 (50.0)
High stress	43 (7.3)	6 (4.0)	0 (0.0)
Very high stress	6 (1.0)	2 (1.3)	0 (0.0)

* 1 (between 2 and 4 h), 2 (between 2 and 4 h), 3 (more than 4 h). ** “Low adherence” (less than 8 points), “good adherence” (between 8 and 10 points), “high adherence” (more than 10 points). *** “Low level” (1 point), “medium level” (2 points), “high level” (3 points). Source: own elaboration.

**Table 2 healthcare-11-02401-t002:** Relationship between academic stress couples (physical exercise as a reference).

Couples	Test Statistic	Standard Error	Standard Test Statistic	*p*	Adjusted Significance *
Very high stress–Low stress	105.209	65.402	1.609	0.061	0.910
Very high stress–Very low stress	139.454	65.310	2.135	0.033	0.491
Very high stress–No stress	184.280	64.364	2.863	0.004	0.063
High stress–Very low stress	101.597	29.800	3.409	<0.001	0.010
High stress–No stress	146.422	27.663	5.293	<0.001	0.000
Medium stress–No stress	75.780	20.270	3.739	<0.001	0.000
Very low stress–No stress	44.826	18.310	2.448	0.014	0.215

Each row tests the null hypothesis that the distributions of Sample 1 and Sample 2 are equal. Asymptotic significance (two-sided tests) is displayed. The significance level is 0.050. * Significance values have been adjusted by Bonferroni correction for various tests. Source: own elaboration.

**Table 3 healthcare-11-02401-t003:** Size and direction of the effect between physical exercise and stress levels.

	Media	Standard Deviation	Hedges’ g
No stress	0.496	0.500	0.796 *
Very low stress	0.319	0.532	0.754 *
Low stress	0.283	0.452	0.455
Medium stress	0.292	0.457	0.475
High stress	0.102	0.102	0.285
Very High stress	0.001	0.000	0.289

* Significant relationship under 0.001 level.

**Table 4 healthcare-11-02401-t004:** Degree of saturation of the items of each factor.

	Factor 1	Factor 2
Fruit consumption	−962	0.007
Olive oil consumption	−0.022	0.99
Consumption of highly processed snacks	0.962	−0.007
Consumption of sugary drinks	0.902	−0.038
Consumption of processed baked goods	0.955	−0.030
I consume natural and/or roasted nuts	−0.022	0.99
Oily fish consumption	−0.022	0.98

Excluded variables: red meat, white meat, white fish, water, legumes, and vegetables. Source: own elaboration.

**Table 5 healthcare-11-02401-t005:** Relationship between type of diet and academic stress.

		Factor 1 (Unhealthy Eating)	Factor 2 (Healthy Eating)	Academic Stress
Spearman’s Rho	Factor 1 (unhealthy eating)	1.00	−0.019	0.700 **
	Factor 2 (healthy eating)	−0.019	1.00	0.076 *
	Academic stress	0.700 **	0.076 *	1.00

** Correlation is significant at the 0.001 level. * Correlation is significant at the 0.05 level. Source: own elaboration.

## Data Availability

Data is unavailable due to privacy.

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
