# Peer review of "Academic Stress in University Students: The Role of Physical Exercise and Nutrition"

_healthcare, 2023, doi:10.3390/healthcare11172401_

Round 1

Reviewer 1 Report

attached

Author Response

Please, see the attachment. Thank you.

Reviewer 2 Report

Manuscript’s Title: Academic stress in university students. The role of physical exercise and nutrition.

A BRIEF SUMMARY 

This review article, entitled “Academic stress in university students. The role of physical exercise and nutrition”, focuses on a pertinent topic. 

The manuscript presents an interesting and always current topic in the field of the interface between psychology and nutrition. The literature review it presents and the final results seem to contribute to a better understanding of the role of physical exercise and the Mediterranean diet in the stress levels of university students. With indications/suggestions for the improvement of stress levels through the consumption of healthy foods and exercise. 

GENERAL CONCEPT COMMENTS

Title and abstract and key words

Title

- The title is appropriate and representative of the main goal of the study, however, instead of "." (leaving the title with two periods) we suggest ":". 

Academic stress in university students: The role of physical exercise and nutrition.

Key Word

- The key words adequately systematize the main aspects and themes addressed in the manuscript. Suggestion: it might be appropriate to add the word "Mediterranean". Suggestion: Indicate the key words in alphabetical order.

Abstract

- The abstract summarizes the manuscript, but it could be improved: some reasons for stress in the university context could be indicated or the generalised indication that there are several reasons could be removed.

I. Introduction

- The introduction is well structured and supports the research questions of the study; 

- Line. 37. “...in which hormonal, cognitive and emotional changes occur.” This part of the sentence appears to be unsupported with reference.

- Line 58: "WHO". Put in full the first time you refer to it (World Health Organisation, WHO). 

- Overall we think that the question of higher consumption of drinks, sugars, snacks and lower consumption of fruit.... in individuals with more stress or the other way round can be problematised? What is the point??

2. Materials and Methods

- Statistic analysis: This topic justifying the analysis decisions could be more condensed.

- Check some mistakes in the text. For example line 140 "body, ."

3. Results 

- The data analysis and reported results aim to address the objectives. 

- Check uniformity and format (e.g, spacing and letter or asterisk legends?) in the notes to the tables.

- "However" is a word often repeated throughout the results and discussion.

4. Discussion

- The discussion held is pertinent and shows reflection on the results found.

5. Limitations and future directions

- If the study focuses on a variable such as stress, it could be interesting to problematise in future studies the "function" of eating certain foods and associate it with emotional states or even the function of emotional regulation through consumption.

6. Conclusions

- The conclusion is clear.

References

References are adequate and current.

Author Response

Please, see the attachment. Thank you.

Round 2

Reviewer 1 Report

The study can be accepted in its present form.